# Removal of organic contamination from wastewater using granular activated carbon modified—Polyethylene glycol: Characterization, kinetics and isotherm study

Leila Choopani[1], Mohammad Mehdi Salehi[1], Hossein Mashhadimoslem[2]*, Mobin Safarzadeh Khosrowshahi[3], Mashallah Rezakazemi[4]*, Ali A. AlHammadi[5,6], Ali Elkamel[2,6], Ali Maleki[1]*

**1** Catalysts and Organic Synthesis Research Laboratory, Department of Chemistry, Iran University of Science and Technology, Tehran, Iran, **2** Chemical Engineering Department, University of Waterloo, Waterloo, Ontario, Canada, **3** Nanotechnology Department, School of Advanced Technologies, Iran University of Science and Technology, Narmak, Tehran, Iran, **4** Faculty of Chemical and Materials Engineering, Shahrood University of Technology, Shahrood, Iran, **5** Center for Catalysis and Separations, Khalifa University, Abu Dhabi, United Arab Emirates, **6** Department of Chemical Engineering, Khalifa University, Abu Dhabi, United Arab Emirates

\* hmashhadimoslem@uwaterloo.ca (HM); mashalah.rezakazemi@gmail.com (MR); maleki@iust.ac.ir (AM)

## Abstract

To effectively remove Diazinon (DZ), Amoxicillin (AMX), and Crystal Violet (CV) from aquatic environments, a novel granular activated carbon (GAC) modified with Polyethylene glycol 600 (PEG) was created and manufactured. The chemical properties were investigated using a variety of characteristic analyses, including FT-IR, XRD, FESEM, and $N_2$ adsorption/desorption. The effectiveness of GAC-PEG's adsorption for the removal of DZ, AMX, and CV was assessed under a variety of conditions, including a pH of 4–9 for the solution, 0.003–0.05 g doses of adsorbent, 50–400 ppm starting concentration, and a reaction time of 5–25 min. For DZ, AMX, and CV adsorption, the maximum adsorption capacity ($Q_{max}$) was 1163.933, 1163.100, and 1150.300 mg g$^{-1}$, respectively. The Langmuir isotherm described all of the data from these adsorption experiments, and the pseudo-second-order well explains all-adsorption kinetics. Most contacts between molecules, electrostatic interactions, π–π interactions, hydrogen bonding, and entrapment in the modified CAG network were used to carry out the DZ, AMX, and CV adsorption on the GAC-PEG. The retrievability of the prepared adsorbent was successfully investigated in studies up to two cycles without loss of adsorption efficiency, and it was shown that it can be efficiently separated.

## Introduction

Clean water is essential for human health and the ecosystem. In many parts of the world, access to clean and safe water is a significant challenge. To be clean and safe, water must be free of all pollutants and harmful factors [1]. Pollutants include chemical, physical, and biological threats

**Data Availability Statement:** All relevant data are within the manuscript and its Supporting Information files.

**Funding:** The author(s) received no specific funding for this work.

**Competing interests:** The authors have declared that no competing interests exist.

to health. Diazinon (DZ), known as dimpylate, is an organophosphorus pesticide (OPPs) frequently employed in urban and agricultural settings. The World Health Organization (WHO) has categorized this substance as Class II, Moderately Hazardous, and its lethal oral dosage, 50% (LD50), for rats, has been determined to be 18 mg kg$^{-1}$ [2]. High residual DZ concentrations (0.41 to 1 g L$^{-1}$) were also discovered in urban waterways and wastewater treatment plant effluents [3]. Antibiotics are considered essential medical medicines, but their release into the environment and aquatic resources has had adverse effects even at modest doses [4]. Semi-synthetic penicillin, known as amoxicillin (AMX), among the most popular in Europe and other parts of the world, has been used alone and in combination with the beta-lactamase inhibitor clavulanic acid since the 1970s [5]. The main characteristics of AMX and clavulanic acid are discussed in this paper, along with the current oral formulations and clinical experience with them [6]. Crystal Violet (CV) is a toxin dye that, according to research, can cause skin irritation and ocular problems in addition to the ability to cause cancer in mammalian cells [7]. The chemical structures of DZ, AMX, and CV are illustrated in Fig 1. [8–10].

Removing such organic contaminants from contaminated water is essential to preserve the environment and the general public's health. Several physicochemical methods, including biological degradation, solvent extraction, irradiation, ion exchange, flocculation, coagulation, chemical oxidation, membrane filtration, adsorption, photocatalysis, and precipitation, have been proposed to remove organic contaminants [11]. Adsorption is recognized as an affordable and practical technology for removing organic contaminants from the environment due to its benefits, including simplicity, reusability, selectivity, technological viability, high efficiency, and low-cost process [12]. However, among the disadvantages of the adsorption process, we can point out the difficult recovery, inefficient adsorption method, and weak interactions between the adsorbent surface and the adsorbed surface [13]. In this method, several adsorbents are used, including carbon-based adsorbents. Carbon-based adsorbents appropriate for gas adsorption include CMS, granular activated carbon (GAC), graphene, carbon nanotubes (CNTs), and Graphitic Carbon Nitride [14–16]. GAC is a synthetic carbon substance with a particular surface area and three-dimensional structure [15]. It contains 87 to 97% carbon, accounting for most of its composition. The variable porosity, well-developed porous structure, simplicity of regeneration, and low cost of synthesis contribute to its excellent applicability. One of the most important properties is sorption capacity, which is strongly influenced by surface functionalization and directly controlled by the pore size distribution [17]. Research on GAC modification towards the adsorption process has become popular nowadays. However, the low desorption rate of lengthy chains counteracts this impact. According to reports, polydispersity is another crucial factor in the adsorption of macromolecules [18]. Increasing molecular weight can change PEG's latent enthalpy and transition

**Fig 1.** Chemical structure of (a) DZ, (b) AMX, and (C) CV.

temperature. Arbuckle et al. [19] demonstrated that the pores' limiting size determines the quantity of molecules adsorbed in the case of polyethylene glycol (PEG) and granular active carbon. Numerous research publications discuss using activated carbon-based PEG for the adsorption-based removal of organic contaminants [20]. Khader et al. [21] used rice husk as a biomass-derived adsorbent and commercial GAC to separate the tartrazine color from an aqueous solution and compared the two. The highest dye concentration was adsorbed in pH = 2 and 5 mg $L^{-1}$ tartrazine, corresponding to removal efficiencies of up to 99.81% for GAC at 0.1 g 50 mL$^{-1}$ and 90.45% for rice husk at 0.2 g 50 mL$^{-1}$. The ideal contact time was discovered after 120 min of rice husk and 60 min of GAC. In another study, Tetracycline antibiotics were removed from an aqueous solution by Xie et al. [22] using a one-step calcination procedure to create a Fe-loaded granular activated carbon catalyst.

However, there hasn't been much investigation on PEG-modified GAC to remove organic contamination from wastewater. In this study, GAC was modified with PEG under an argon atmosphere to produce GAC-PEG. The extraordinary structural characteristics of GAC-PEG, such as the incredible morphology of cavities, exceptional qualities of high surface area, and porous structure, were assessed by various analyses such as FESEM, $N_2$ ads/des, XRD, and FT-IR. A maximum adsorption capacity was reported for DZ, AMX, and CV adsorption. A Langmuir and Freundlich isotherm were investigated. A pseudo-first-order and a pseudo-second-order model was developed to describe the adsorption kinetics. As a result, DZ, AMX, and CV achieved their maximum adsorption capability using GAC-PEG.

## Experimental

### Materials and instruments

As shown in Table 1, the materials used in the DZ, AMX, and CV removal processes and the adsorption preparation procedure were from reputed Merck and Sigma Aldrich companies. GAC adsorbent has QC/QA certifications from their manufacturer. In addition, the information about the instruments used for characterization has been compiled in Table 2.

### The GAC-PEG preparation

**Preparation of GAC-PEG adsorbent.** In preparing modified GAC, this black powder was first sieved to a size between 30 and 40 meshes, cleaned repeatedly with distilled (DI) water (2 ppm) to remove contaminants, and then dried for an entire night at 100°C to remove moisture [23]. PEG 600 (2 mL, 2.26 g) and 2 g GAC were combined and agitated in 200 mL of DI water without additional heat. The mixture was then heated for 2 h in an argon-filled furnace to 60°C (heating rate: 3°C min$^{-1}$). The finished product was then cooled to 25°C, rinsed with 1 M HCl to remove any surplus reactants, and filtered through 100 ml of DI water until

**Table 1. Mark and pureness of all materials applied in this research.**

| Material | Brand and Purity |
|---|---|
| Granular Activated Carbon (GAC) | Jacobi Co. was purchased for the carbon-based adsorbent |
| Hydrochloric acid (HCl) | Merck (37.0%) |
| Sodium Hydroxide (NaOH) | Reagent grade, ≥98%, pellets (anhydrous), Sigma-Aldrich |
| Poly Ethylene Glycol (PEG) 600 | Sigma Aldrich |
| Ethanol | Sigma Aldrich (96.0%) |
| Amoxicillin (AMX) | Amoxicillin, 95.0–102.0% anhydrous basis |
| Crystal violet (CV) | ACS reagent, ≥90.0% anhydrous basis |
| Diazinon (DZ) | Sigma-Aldrich, Cas.no (333-41-5) |

**Table 2. The type of applied devices in this work.**

| Instrument | Brand and Model |
|---|---|
| BET | 2020 ASAP$^{TM}$ micropolitics |
| XRD | DRON-8 X-ray diffractometer |
| FT-IR | Shimadzu FT-IR-8400S |
| FE-SEM | Hitachi S-5200 and ZEISS SIGMA |
| Oven | Genlab Ltd |
| UV-vis (solid phase) | Shimadzu-UV-2550/220v |
| Thermometer | Fluke (572–2 infrared) |
| Filter paper | Whatman (grade 602h, Particle retention < 2μm) |
| pH-meter | PHS-3C pH |
| TDS-32 | China |

the pH of the solution was about 5. The amended sample was then dried for 12 h at 80°C. The GAC-PEG adsorbent stands for modified granular activated carbon. The pathway preparation of GAC-PEG is illustrated in Fig 2.

**Batch adsorption experiments.** DZ, AMX, and CV removal were used to study the batch adsorption tests of the GAC-PEG adsorbent in 10 mL of aqueous solution. To investigate the impact of the pH, adsorbent dosage, initial concentration of the DZ, AMX, and CV, and reaction time, experimental successions have been carried out at a pH range of 4–9, 0.003–0.05 g adsorbent dosages, 50–400 ppm initial concentration, and a reaction period of 5–25 min. The pH of the reaction solution was maintained between 4–9 using the NaOH and HCl solutions (0.1 mol L$^{-1}$). UV-vis spectroscopy tracked the DZ, AMX, and CV concentrations in an aqueous solution. Using Eqs 1 and 2, the removal efficiency percentage for each experiment and the adsorption capacity (Qe) were computed.

$$\% \, Adsorption = \left( \frac{Ci - Ce}{Ci} \right) \times 100 \qquad (1)$$

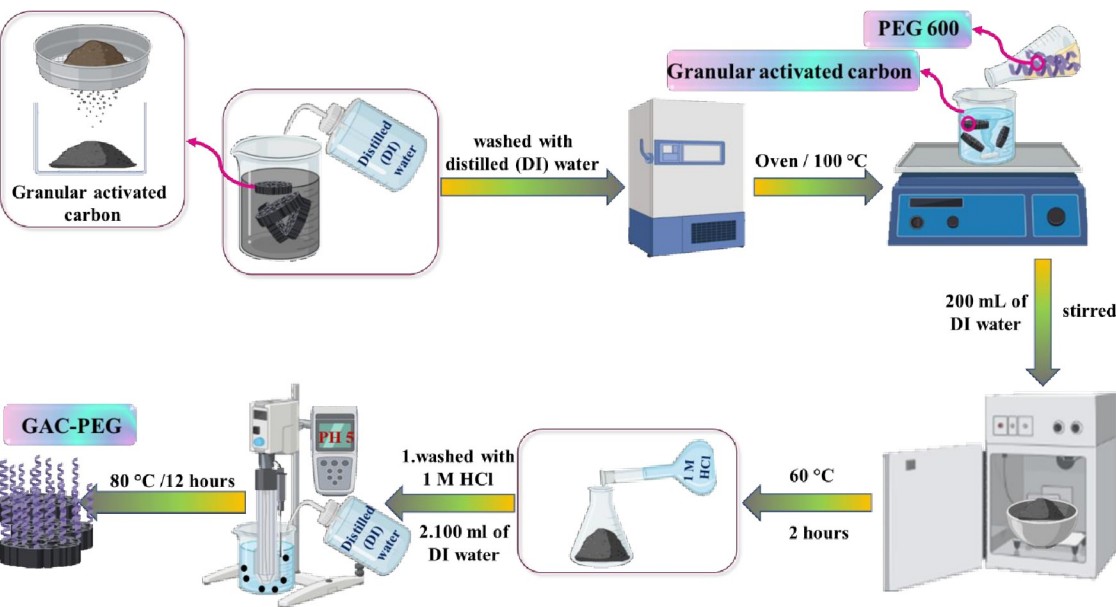

**Fig 2. Schematic illustration of GAC-PEG preparation.**

$$Qe = \left(\frac{Ci - Ce}{m}\right) \times V \tag{2}$$

Where m (g) stands for the applied adsorbent's mass, $C_i$ (ppm) for the solution's initial concentration of metal ions, $C_e$ for each experiment's residual metal ions concentration, and V (L) for the solution's volume.

### Regeneration and reusability

The GAC-PEG adsorbent was introduced to an HCl solution at various pH levels and shaken at room temperature for desorption to assess the adsorbent's reusability. The adsorbent was then collected and thoroughly cleaned with distilled water. Consecutive adsorption-desorption trials were conducted to examine the adsorbent's reusability. Eq (3) was applied to calculate the desorption percentage (D%) as follows:

$$D(\%) = \frac{A}{B} \times 100 \tag{3}$$

B (mg) is the amount of pollutants sorbed on the GAC-PEG sorbent, and A (mg) is the amount of contaminants desorbed in the washing solution.

## Results

Removing organic pollutants from aquatic areas is crucial due to their destructive effect on humans and other living beings. GAC-PEG adsorbent was employed to eliminate DZ, AMX, and CV from contaminated water. Compared to pure GAC, employing GAC-PEG in water treatment may enhance pollutant removal efficiency. The GAC-PEG surface groups and porosity morphology that are accessible for adsorption and their affinity for DZ, AMX, and CV contaminants can both increase with the addition of PEG. Furthermore, PEG's hydrophilicity can enhance the stability and dispersion of GAC particles in water.

### Material characterization

According to Fig 3A, FT-IR spectra are employed to study the chemical structure of prepared materials. Peaks in the FT-IR spectra of GAC at 3430 and 1579 cm$^{-1}$, respectively, suggest the

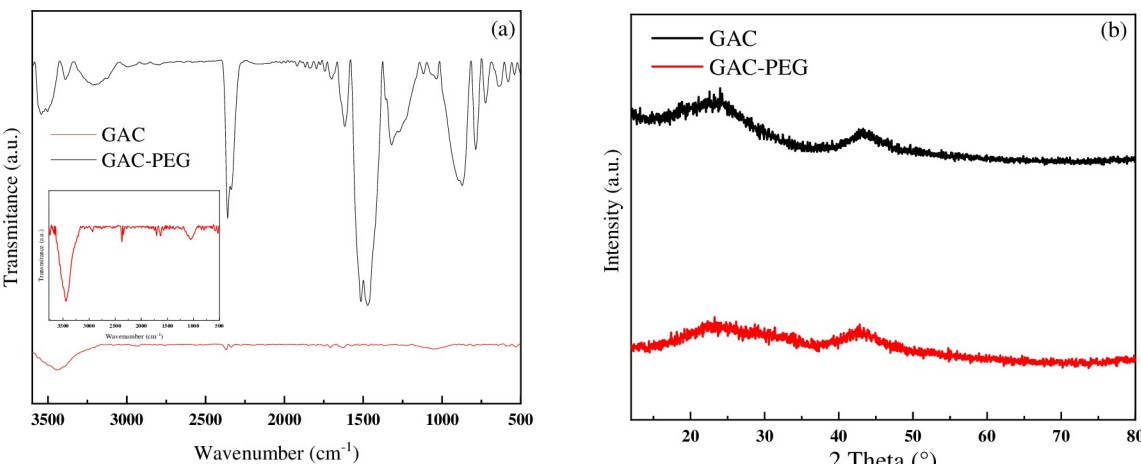

**Fig 3.** (a) FT-IR spectra of GAC, and GAC-PEG (inset: GAC) and (b) XRD pattern of GAC and GAC-PEG.

presence of the O-H group and the C = C group [23]. A peak at roughly 2355 cm$^{-1}$ indicates that the C = C (alkyne) stretching vibration is present. Several bands in the GAC-PEG spectrum, from 1050 to 1150 cm$^{-1}$, are attributed to ether group stretching. From 2850 to 3000 cm$^{-1}$, distinctive alkyl (R- CH$_2$) stretching modes were seen. Additionally, it was noted that the O-H and C = C groups had absorption ranging from 3000–3600 and 1550 cm$^{-1}$, respectively [23]. As a result, the presence of some of these peaks in the modified GAC confirms the tendency to be hydrophilicity due to the presence of PEG. It seems that during the functionalization process, the gradual breaking of the chains has led to the increase of surface functional groups.

X-ray diffraction (XRD) study was used to investigate the crystallinity of the GAC and GAC-PEG adsorbent (Fig 3B). The XRD patterns of GAC and GAC-PEG have firm, recognizable peaks at 2θ = 22˚ (0 0 2), and 2θ = 42˚ (1 0 1), respectively, according to the data mentioned in the articles [23]. Since a carbon plate has randomly stacked them, these peaks imply that the modified GAC is amorphous-graphitic carbon that can be employed to create an adsorption gap.

Using the FESEM images, the morphology, size, shape homogeneity, particle size distribution, and probable agglomerations of the samples were all evaluated and elucidated. According to Fig 4A–4C, the FESEM micrographs of the GAC were captured at 700 and 30 μm magnifications. As can be seen, the pore shape was incomplete, and the external surface of GAC was smooth with no apparent holes. According to the GAC-PEG pore structure (Table 3), PEG modification causes the pores' sizes to shrink. Moreover, the samples' pores have a relatively homogeneous distribution Fig 4D–4F. Even though there seem to be interconnected pores in the modified structure, the excessive presence of functional groups has led to the destruction of pore walls and the blocking of micropores. As a result, according to the N$_2$ ads/des, the specific surface area of the modified sample has decreased (910 m$^2$ g$^{-1}$) [23].

It is well known that the quantity of adsorption sites considerably affects the adsorption capability of porous materials. Nitrogen gas adsorption/desorption studies were conducted to assess the produced samples' porous properties (shown in Fig 5A). Table 3 summarizes the instances' specific textural attributes. According to the IUPAC classification, the isotherms

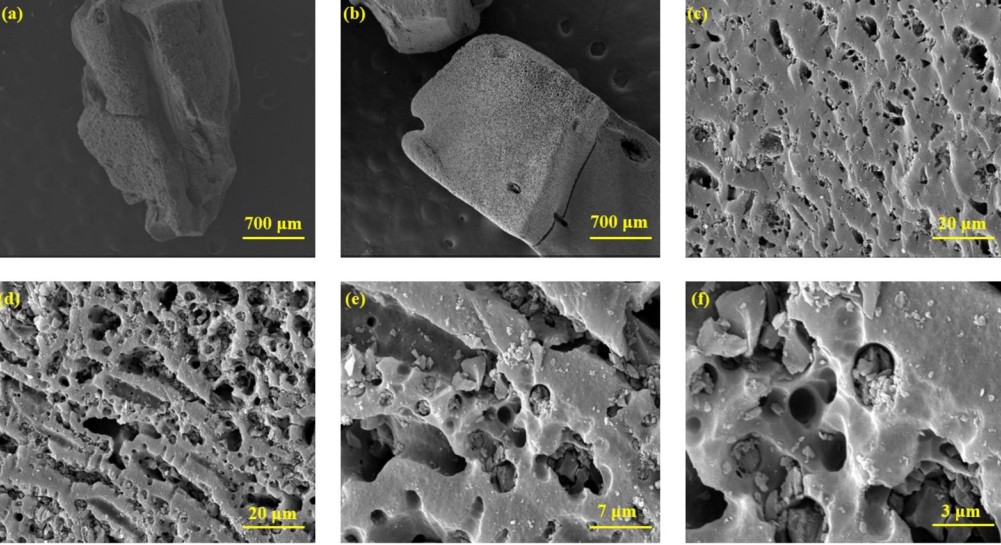

**Fig 4.** (a-c) FESEM micrographs of GAC and (d-f) GAC-PEG.

**Table 3. Specific textural attributes of GAC-Pure and GAC-PEG samples.**

| Sample name | Specific surface area ($m^2 \, g^{-1}$) | Average pore diameter (nm) | Total pore volume ($cm^3 \, g^{-1}$) | Mesoporous volume ($cm^3 \, g^{-1}$) | Microporous volume ($cm^3 \, g^{-1}$) |
|---|---|---|---|---|---|
| GAC-Pure | 921 | 1.80 | 0.41 | 0.05 | 0.36 |
| GAC-PEG | 910 | 1.75 | 0.39 | 0.06 | 0.33 |

appeared to be type I isotherms because they were horizontal over a wide pressure range and did not have any obvious hysteresis loops. All of the products' nitrogen adsorption capabilities rise quickly in the low relative pressure region ($0 < P/P_0 < 0.2$), then increase continuously with rising relative pressure until reaching a nearly constant value at about $P/P_0 = 0.95$. As a result, the modified GAC had microporous surfaces with pore-size distribution curves below 2 nm (Fig 5B). The samples displayed relatively modest $N_2$ adsorption capabilities, as was expected, given that the porous structure of these materials is almost entirely composed of micropores. The diffusion of $N_2$ molecules into micropores is sluggish at cryogenic temperatures. The value of surface area computed from the $N_2$ adsorption isotherm for GAC-PEG became lower than the GAC ($910 \, m^2 \, g^{-1}$) because of the smaller average pore diameter (1.75 nm) [23]. The distribution peaks for the GAC and GAC-PEG samples were centered at 1.8 nm using the Barrett-Joyner-Halenda (BJH) method.

## Optimization of the effective parameters

The removal of organic pollutants from aquatic environments using solid-sorbents was influenced by the pH of the solution, the length of the sorption equilibrium, the dose of sorbent, and, eventually, the initial concentration of DZ, AMX, and CV on sorbents. The efficiency of clearing organic contamination can be significantly improved by changing these factors.

## Solution pH

The adsorption process is substantially impacted by pH changes in the solution. The electrostatic interaction between the ions' adsorption surfaces and the solution's pH frequently determines the outcome. The chemical solution and the adsorbents' surface binding sites can both be impacted by the pH of the solution. The current study investigated how pH impacted DZ,

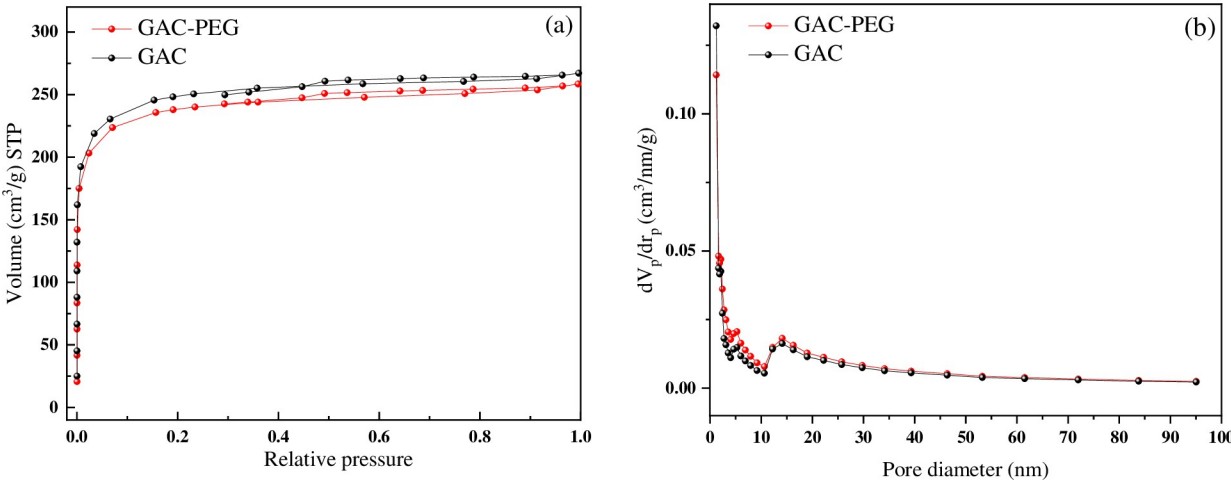

**Fig 5.** (a) The nitrogen adsorption/desorption, and (b) pore size distribution of GAC and GAC-PEG.

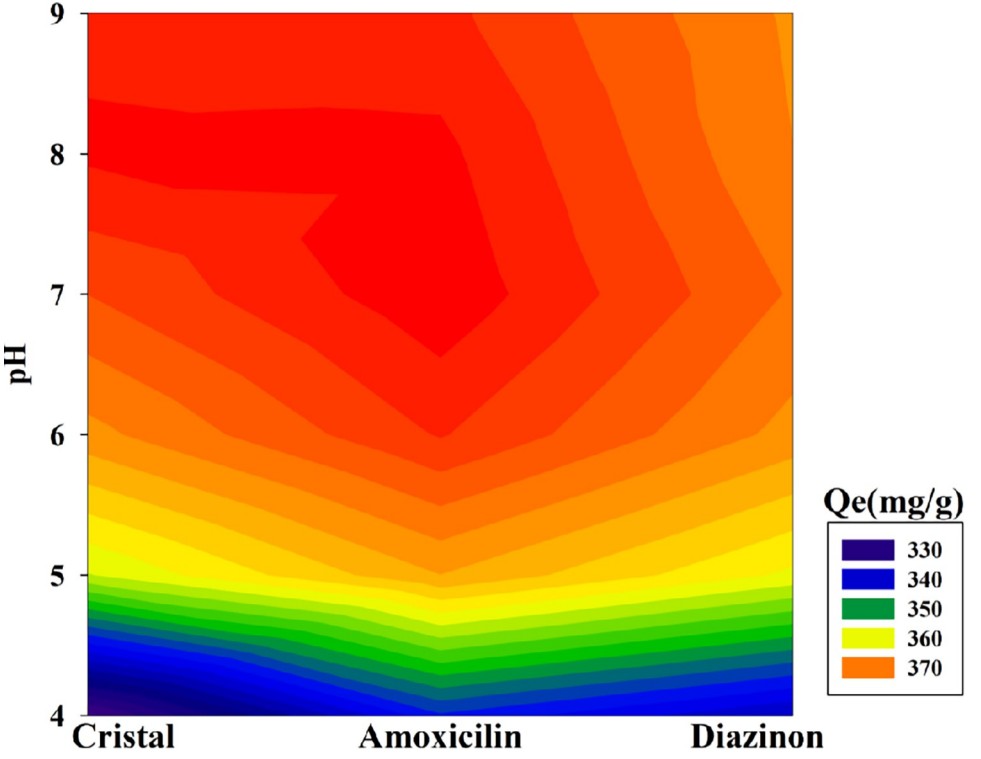

**Fig 6. pH changes on the absorption of DZ, AMX, and CV.**

AMX, and CV absorption by the GAC-PEG adsorbent, whose schematic is shown in Fig 6. Electrostatic attraction between positive charges occurs at acidic pH because of the greater concentration of $H^+$ ions in the solution and the protonation of functional groups (hydroxyl) on the surface of the adsorbent. As a result, the adsorbent's DZ, AMX, and CV molecules exhibit less electrostatic attraction to one another, lowering adsorption capacity. The DZ, AMX, and CV molecules' adsorption capacities were calculated with increasing pH. The adsorption capacity for DZ and AMX begins to rise when the pH reaches between 4 and 7; for CV, it rises when the pH reaches between 4 and 8. At neutral pH (7) and (8), the maximal adsorption capacities for DZ, AMX, and CV were 369.78, 376.54, and 376.38 mg $g^{-1}$, respectively, whose schematic is shown in Fig 7A. Due to the positive charge on the adsorbent surfaces at acidic pH, DZ, AMX, and CV also exist in acidic pH as cations. The primary mechanisms that bind DZ, AMX, and CV to GAC-PEG adsorbent include hydrogen bonds, electrostatic interactions, and π–π interactions. It should be stressed that in terms of their impact on adsorbent adsorption, other critical factors like time and the solution's initial concentration, including DZ, AMX, and CV, are fixed. Other important parameters, such as time and the starting concentration of the solution comprising DZ, AMX, and CV, were thought to be fixed at this point in terms of their effects on the adsorbent's absorption. 10 ml of a solution containing DZ, AMX, and CV at various pH (4, 5, 6, 7, 8, and 9) were tested for 25 min on 0.005 g fixed adsorbent with an initial concentration of 200 ppm at room temperature.

## Adsorbent dosage

This stage involved investigating the impact of GAC-PEG adsorbent's impact on DZ, AMX, and CV absorption; the resulting diagram is given in Fig 7B. The adsorption capacity of DZ,

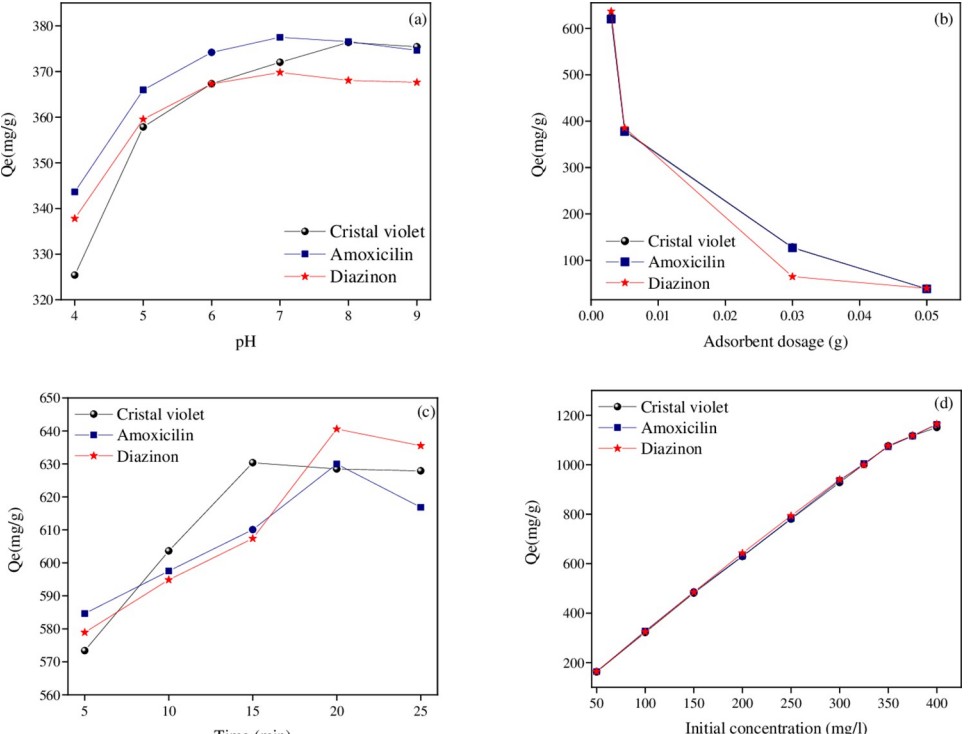

**Fig 7.** (a) Effects of solution pH (4.0–9.0), (b) Adsorbent dosage (0.003–0.05 g), (c) Contact time (5–25 min), and (d) Initial concentration (50–400 mg L$^{-1}$).

AMX, and CV increases as the amount of adsorbent is reduced. This increase in capacity is often brought about by a rise in the amount of pollutant that can be absorbed per unit mass of the sorbent. However, the number of sites or surface-active sites necessary to complete the adsorption process decreases as the amount of adsorbent increases as more adsorbent molecules aggregate and interact. With 0.003 g of adsorbent, DZ, AMX, and CV have maximum absorption capacities of 636.766, 620.1, and 623.4 mg g$^{-1}$. In this stage of optimization, different amounts of the adsorbent (0.003, 0.005, 0.03, and 0.05 g) were added to a solution of DZ, AMX, and CV in 10 ml at fixed initial concentrations of 200 ppm, ideal pH conditions of 7 (from the previous stage for DZ, AMX, and CV), fixed 25 min, and ambient temperature.

## Contact time

The contact time is a critical factor in the adsorption process for proper adsorbent efficiency for real-world applications, determining the binding speed, and determining the best period, time for the complete removal of DZ, AMX, and CV contaminants. To explore the adsorption of the tested pollutant by GAC-PEG adsorbent, the optimal amount of adsorbent and the effect of contact time on pollutant adsorption at pH were examined between 5 and 25 min for DZ, AMX, and CV. According to Fig 7C, the adsorption capacity rises with increased adsorbent contact time and is, respectively, 640.73, 629.97, and 630.40 mg g$^{-1}$ for solutions containing DZ, AMX up to the first 20 min, and CV up to the first 15 min. However, with an increase in contact time, there is no appreciable difference in the growth in absorption capacity. The optimal amount of adsorbent, ideal pH conditions, a fixed concentration of 200 ppm, and 10 ml of the solution containing DZ, AMX, and CV at various times (5, 10, 15, 20, and 25 min) were all taken into account during the time optimization step for the adsorbent.

## The initial concentration

To determine the effect of the beginning concentration on the absorption rate of DZ, AMX, and CV, the absorption process was conducted under the optimal conditions of adsorbent quantity, duration, pH, and concentration between 50 and 400 mg $L^{-1}$. As shown in Fig 7D, the results of these studies show a direct relationship between the pollutant's initial concentration and absorption capacity. By measuring 10 ml of the pollutant solution at various concentrations (50–400 mg $L^{-1}$) with 0.003 g of adsorbent for 15 min at room temperature and 20 min at the ideal pH for DZ and AMX, the optimization of the pollutant's initial concentration was examined in the final step. As a result, by raising the concentration of DZ, AMX, and CV solution from 50 to 400 mg $L^{-1}$, the GAC-PEG adsorbent's adsorption capacity was enhanced to 1163.933, 1163.1, and 1150.3 mg $g^{-1}$, respectively.

## Adsorption isotherms and kinetics studies

The surface properties of the adsorbent are described using models and equations of adsorption equilibrium isotherms, which are also used to describe experimental data and give an overview of the surface adsorption process [24]. The design of adsorption systems and the characterization of the link between the concentration of the adsorbed substance and the adsorption capacity of an adsorbent heavily rely on isotherms. In the current study, the analysis of experimental data as well as Langmuir and Freundlich equilibrium isotherm models were used to explain the relationship between the concentration of adsorbed material and the adsorption capacity to look into the reaction between the adsorbed material and the adsorbent. The Langmuir isotherm model assumes that a single layer of uniform (homogeneous) adsorbent material will be adsorbed with the same energy on all surfaces of the sorbent; it means that the adsorption will take place only in this homogeneous location and will not involve any molecular reactions. There are adsorbent and adsorbent materials. Eq (4) represents its linear equation [25].

$$\frac{C_e}{q_e} = \frac{1}{K_L Q_{max}} + \frac{1}{q_{max}} C_e \tag{4}$$

Here, $Q_e$ represents the maximum amount of DZ, AMX, and CV that may bind to GAC-PEG, while $k_L$ stands for the Langmuir constant (L $mg^{-1}$). The amount of adsorbate at equilibrium time (mg $g^{-1}$), and Ce, the equilibrium concentration of the adsorbed material (mg $L^{-1}$); the values of the coefficients $Q_e$ and kL are calculated, respectively, using the width from the origin and the slope of the linear graph of Ce/ $Q_e$ against Ce.

The experimental equation for the Freundlich isotherm, Eq (5), is based on the adsorbed substance's multi-layer and heterogeneous adsorption on the adsorbent, in contrast to the Langmuir model.

$$ln\, q_e = ln\, K_F + \frac{1}{n} ln\, C_e\, ln\, q_e \tag{5}$$

In this equation (mg $g^{-1}$), $Q_e$ is the equilibrium adsorption capacity, Ce is the equilibrium concentration of the adsorbent, $K_F$ (mg $g^{-1}$) is the relative adsorption capacity (Freundlich constant), and n indicates the tendency of the adsorbed molecules to be adsorbed on the adsorbent. This model's values of n less than one imply weak absorption. Fig 8A and 8B. It displays diagrams for the Freundlich and Langmuir isotherm models. Table 4 shows all of the data from these three isotherms. According to the matching correlation coefficient ($R^2$) of isotherm models, the Langmuir isotherm was found to be more following the experimental data than the Freundlich isotherm ($R^2_{DZ} = 0.9982$, $R^2_{AMX} = 0.9942$, and $R^2_{CV} = 0.9931$). Table 5

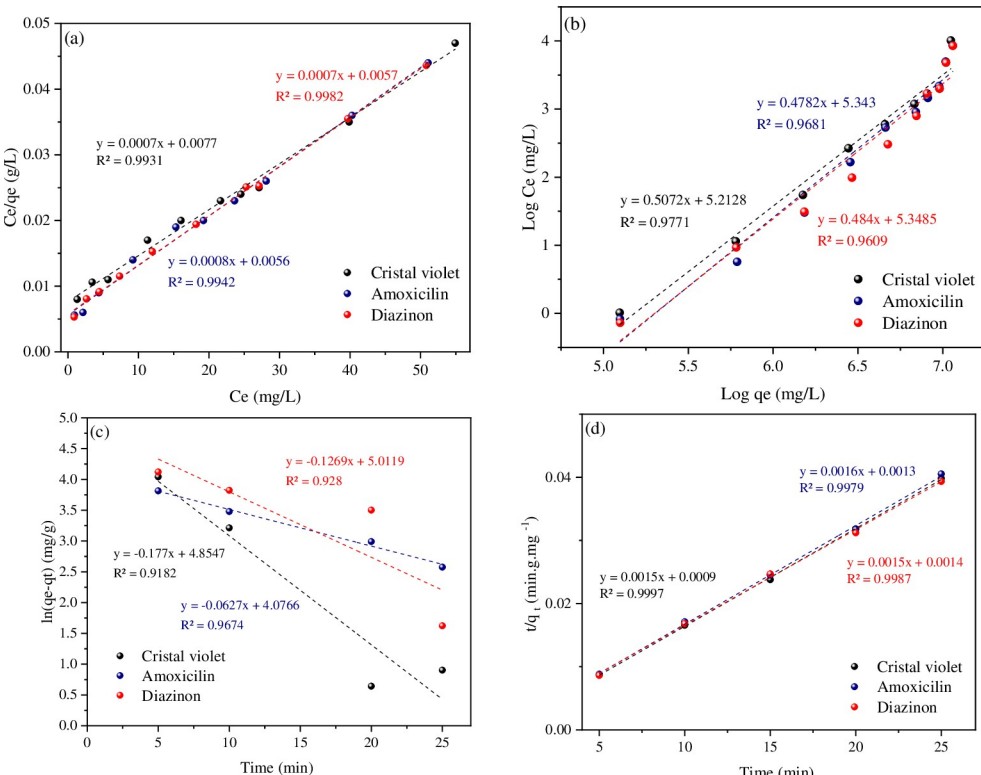

**Fig 8.** (a) The Langmuir, (b) Freundlich, (c) Pseudo-first-order and (d) pseudo-second-order models.

compares the maximal DZ, AMX, and CV adsorptive capability of the GAC-PEG to those of other adsorbents mentioned in earlier studies. Compared to several other adsorbents that have been previously reported, the created GAC-PEG adsorbent showed high $Q_{max}$. The different physicochemical properties of the made adsorbent systems, including a huge surface area and plenty of reactive adsorption sites, such as hydroxyl groups covering the surface, effectively adsorb OPPs, antibiotics, and toxic dyes from wastewater pollutants.

**Table 4. Isotherm, kinetic constants, and correlation coefficients for DZ, AMX, and CV adsorption on the GAC-PEG adsorbent.**

| Model | Parameters | CV | AMX | DZ |
|-------|-----------|----|----|----|
| **Isotherm** | Freundlich | $K_F$, mg g$^{-1}$ | 183.5 | 209.1 | 209.1 |
| | | n | 1.97 | 2.09 | 2.09 |
| | | $R^2$ | 0.9771 | 0.9681 | 0.9609 |
| | Langmuir | $Q_{max}$ (mg g$^{-1}$) | 1428.57 | 1250.00 | 1428.57 |
| | | $K_L$ (L mg$^{-1}$) | 0.091 | 0.143 | 0.123 |
| | | $R^2$ | 0.9931 | 0.9942 | 0.9982 |
| **Kinetics** | Pseudo-first-order | $k_1$ (min$^{-1}$) | 0.1777 | 0.0627 | 0.0920 |
| | | $Q_{e, \text{experimental}}$ (mg g$^{-1}$) | 630.40 | 629.97 | 640.73 |
| | | $Q_{e \text{ calculated}}$ (mg g$^{-1}$) | 133.66 | 58.94 | 59.86 |
| | | $R^2$ | 0.9182 | 0.9674 | 0.928 |
| | Pseudo-second-order | $k_2$ (g mg$^{-1}$ min$^{-1}$) | 0.00250 | 0.00197 | 0.00161 |
| | | $Q_{e, \text{experimental}}$ (mg g$^{-1}$) | 630.40 | 629.97 | 640.73 |
| | | $Q_{e \text{ calculated}}$ (mg g$^{-1}$) | 666.67 | 625.00 | 666.67 |

**Table 5. Evaluation of the maximum adsorptive capability of GAC-PEG compared to the previously reported studies.**

| Adsorbent | $Q_{max}$ (mg.g$^{-1}$) for CV | Ref. | Adsorbent | $Q_{max}$ (mg.g$^{-1}$) for AMX | Ref. | Adsorbent | $Q_{max}$ (mg.g$^{-1}$) for DZ | Ref. |
|---|---|---|---|---|---|---|---|---|
| AC | 32.258 | [26] | AC Prepared (GSA) | 2.28 | [28] | PAAC composite | 1.48 | [30] |
| ACL/Fe$_3$O$_4$ | 35.3 | [27] | Av-S-Ac | 28.86 | [29] | Nano-MnO2/PAC composite | 11.512 | [31] |
| PLAC | 70.32 | [32] | Phosphoric acid-activated carbon (PAC) | 51.8 | [33] | AC–ZnO nanocomposite | 44 | [34] |
| Merck activated carbon | 84.11 | [35] | CAC | 67.7 | [36] | AC | 67.3 | [37] |
| AC-COOH | 120 | [38] | AC derived from Jujube nuts | 77.78 | [39] | Activated Watermelon Rind | 68.15 | [40] |
| GO-AC | 147 | [41] | PAC | 80.41 | [42] | (AC) derived from walnut shell | 169.49 | [43] |
| CoFe$_2$O$_4$/AC | 184.2 | [44] | AC from the coconut shells | 192.31 | [45] | (Fe-AC) | 191.1 | [37] |
| ACMAS | 235.7 | [46] | LS–AC–SG nanocomposite | 249 | [47] | KAsAC | 234.25 | [48] |
| (PACK) | 497.51 | [49] | GAC | 312.5 | [50] | (NAC) | 250 | [51] |
| **GAC** | **438.32** | **This Work** | **GAC** | **564.87** | **This Work** | **GAC** | **516.21** | **This Work** |
| **GAC-PEG** | **1163.933** | **This Work** | **GAC-PEG** | **1163.100** | **This Work** | **GAC-PEG** | **1150.300** | **This Work** |

Kinetic equations are used to analyze the factors influencing reaction speed or to explain the behavior of molecules of an adsorbate per unit of time. The factors impacting the reaction speed of the DZ, AMX, and CV adsorption processes on GAC-PEG adsorbent were investigated in the current study using the most helpful pseudo-first-order and pseudo-second-order kinetic models. c Eqs (6 and 7) are the first and second-order pseudo-linear kinetic equations, respectively.

$$\text{Log}(Q_e - Q_t) = \text{Log}Q_e - \frac{k_1}{2.303}t \qquad (6)$$

$$\frac{t}{Q_t} = \frac{1}{k_2 Q_e^2} + \frac{1}{Q_e}t \qquad (7)$$

where $Q_e$ and $k_1$ are, respectively, the width from the origin and the slope of the linear graph ln $(Q_e−Q_t)$ versus t, and $k_2$ is the quasi-quadratic reaction constant in terms of g mg$^{-1}$ min$^{-1}$, the adsorption capacity at equilibrium time, and time t in terms of mg g$^{-1}$. The slope and width from the origin of the linear graph of t/qt against t in Eq 7 can be used to calculate the values of $Q_e$ and $k_2$. Table 4 Also includes the kinetic parameter values for the GAC-PEG adsorbent used in the DZ, AMX, and CV adsorption processes. According to these results, Fig 8C and 8D, it can be concluded that the adsorption process follows the pseudo-second-order model and that chemical absorption was the limiting step in the absorption process. As can be seen in Table 4, the correlation coefficient for each of the two studied kinetic models is equal to 0.9987, 0.9979, and 0.9997, respectively.

## Recovery and reusability

In sustainability, which aims to decrease waste and save resources, recovery and reusability are two essential principles. It has come to light recently due to growing worries about the destruction of the environment and the depletion of natural resources. Implementing recycling and reusability measures to reduce waste and conserve resources is economically significant.

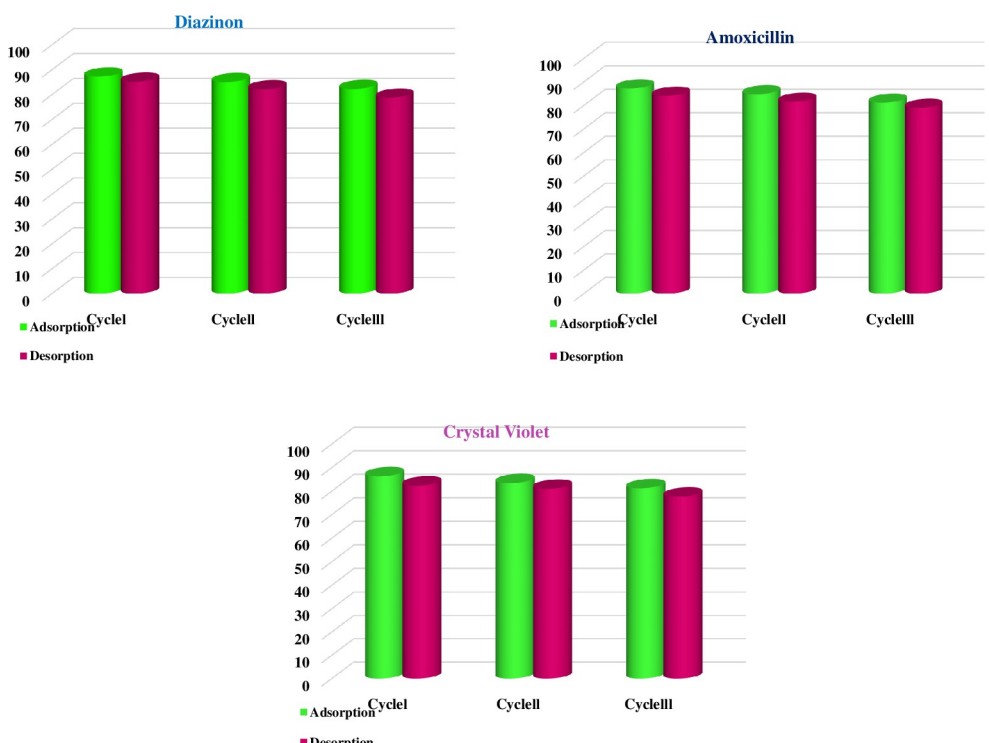

**Fig 9. Reusability of the GAC-PEG adsorption/desorption percentages of DZ, AMX, and CV during three cycles.**

Adsorbents with these characteristics thereby lessen the difficulties associated with adsorbent disposal and the production of new adsorbents. Three successive cycles of adsorption and desorption investigations were performed to assess the recoverability and regeneration of the GAC-PEG adsorbent. The desorption method was performed by immersing the adsorbent loaded with DZ and AMX in ethanol (4 hours) and CV in 0.1 mol.L$^{-1}$ HCl for four hours at room temperature. Following the release of DZ, AMX, and CV from the mixture, the sorbent was filtered out, thoroughly rinsed with distilled water and ethanol, and dried for use in further absorption/desorption experiments. According to Fig 9, after three cycles, the adsorption percentage for DZ and 87.295% to 82.33%, AMX from 81.21 to 87.2325%, and CV and 87.295 to 82.33%, and the percentage of disposal from 85.224 to 78.82%, 84.13 to 79.02% and 82.34 to 877.64% for DZ, AMX, and CV, decreased respectively. The adsorbent system set up to eliminate DZ, AMX, and CV has sorbed for three consecutive periods, yet it is still effective and stable. The lower absorption effectiveness could be due to the loss of some adsorbent during washing. One of the crucial characteristics of a typical adsorbent that was examined is adsorption repeatability. The main goals of the renewable evaluation were to choose an affordable adsorbent and minimize any potential environmental effects following its disposal. Under ideal circumstances, the GAC-PEG adsorbent was used to test the repeatability of the adsorption process using the DZ, AMX, and CV methods.

## Adsorption mechanisms

The exceptional physicochemical properties of the GAC-PEG adsorbent system can also be employed to describe the adsorption mechanism in the adsorption processes of DZ, AMX, and CV. The proposed mechanism for the adsorption of DZ, AMX, and CV from water by GAC-PEG in Fig 10. It is presented as follows: I) The O-H groups present on the surface of

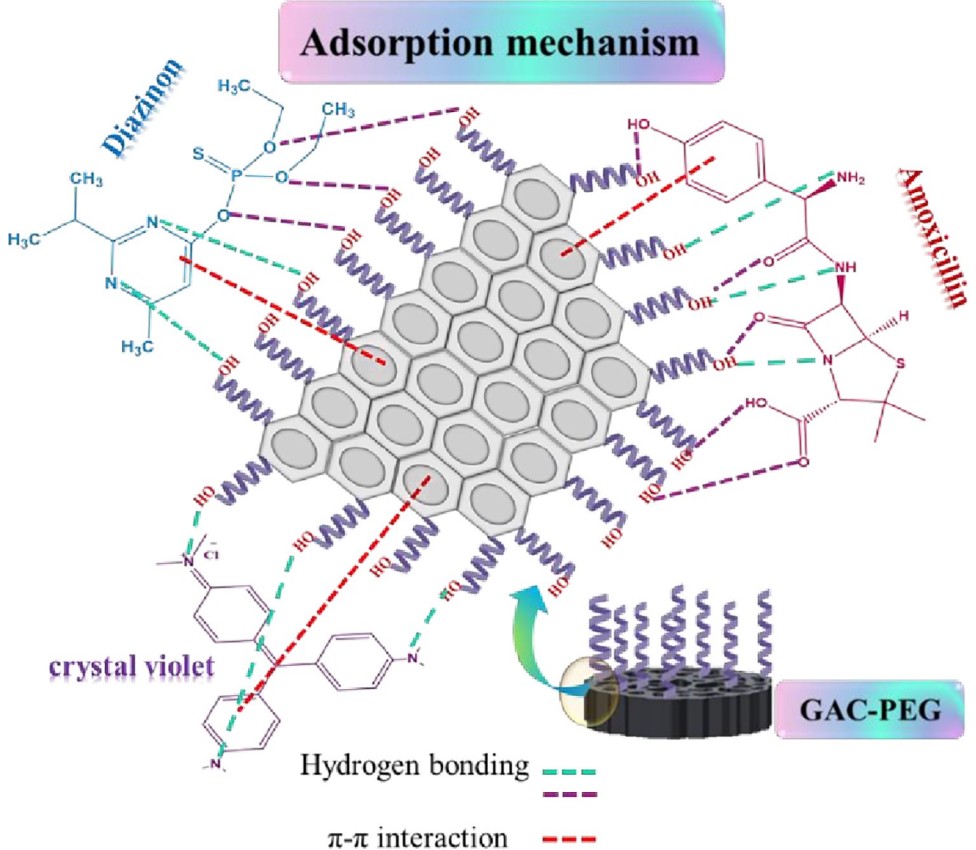

**Fig 10. The suggested mechanism for adsorption of DZ, AMX, and CV by the GAC-PEG adsorbent.**

GAC-PEG form hydrogen bonds with the functional groups present in the pollutant (NH, OH). II) The π-π interaction between the GAC-PEG structure and the aromatic structure in DZ, AMX, and CV molecules. III) Diffusion and transfer of DZ, AMX, and CV pollutants into the pores and pores of GAC-PEG. All these mechanisms effectively contributed to the absorption performance [52, 53].

## Conclusions

As mentioned, in this project, due to the harmful effect of organic pollutants on humans and other living organisms, granular activated carbon (GAC) that has been treated with polyethylene glycol (PEG) was created to serve as an effective adsorbent system from contaminated water. GAC-PEG adsorbent, made as an environmentally friendly multifunctional adsorbent to remove DZ and AMX from the aqueous medium at pH 7–20 min, and CV at pH 8–15 min with the adsorbent value of 0.003 and concentration 400. $N_2$ adsorption/desorption, XRD, FE-SEM, and FT-IR analysis were performed to identify the structure characteristics of the GAC-PEG adsorbent. According to the results of adsorption and desorption, the fine GAC-PEG offers a surface area of 910 $m^2\ g^{-1}$. Additionally, it had a total pore volume of 0.39 $cm^3\ g^{-1}$, an average pore diameter of 1.75 nm, a mesoporous volume of 0.06 $cm^3\ g^{-1}$, and a micropore volume of 0.33 $m^3\ g^{-1}$. OH, groups on the GAC-PEG structure create hydrogen bonds with DZ, AMX, and CV, and their targets from the aqueous environment. The experimental data analysis and the adsorption process description were performed using Langmuir and

Freundlich isotherm models to explain the relationship between the concentration of adsorbed substances and the adsorption capacity. The matching correlation coefficient ($R^2$) of isotherm models showed that the Langmuir isotherm followed the experimental data more than the Freundlich isotherm. This adsorbent showed more $Q_{max}$ compared to the other adsorbents. The maximum adsorption capacity of DZ>AMX>CV was 1163.933, 1163.1, and 1150.3 mg g$^{-1}$, respectively, using GAC-PEG adsorbent. The parameters that affected the reaction speed of the DZ, AMX, and CV adsorption processes on GAC-PEG adsorbent were assayed using the pseudo-first and second-order kinetic models, and the obtained data showed that the reaction kinetics is the pseudo-second-order model and that chemical adsorption was the limiting step in the absorption process. The GAC-PEG adsorbent is an adaptable, environmentally friendly system with exceptional qualities, including various active adsorbent sites, high specific surface area, and retrievability. To improve and make the pollutant Adsorption process by the adsorbent more effective, modifications can be made to the composite mentioned in this project. (I) It can be magnetized, which makes it possible to create a surface adsorption process without the need for smoothing after purifying the pollutant solution. In fact, after the completion of the purification process, the adsorbent can be easily separated with the help of an external magnetic field, which reduces the time and increases the quality of the purification process. (II) The adsorbent can be designed to act selectively to remove a specific pollutant and directly remove that. (III) Using two-dimensional (2D) materials such as MOF can be modified to increase its adsorption capacity for these pollutants.

## Supporting information

**S1 File. The support information (SI) file presents the standard calibration curve for determining diazinon, Amoxicillin, and Crystal violet concentration in an aqueous solution as S1–S3 Figs.**
(DOCX)

## Acknowledgments

All authors are grateful for support from the respective universities (Iran University of Science and Technology; University of Waterloo; Khalifa University, and Shahrood University of Technology).

## Author Contributions

**Conceptualization:** Leila Choopani, Hossein Mashhadimoslem.

**Data curation:** Leila Choopani.

**Formal analysis:** Leila Choopani, Mobin Safarzadeh Khosrowshahi.

**Funding acquisition:** Ali Maleki.

**Investigation:** Mohammad Mehdi Salehi, Mobin Safarzadeh Khosrowshahi.

**Methodology:** Leila Choopani.

**Project administration:** Hossein Mashhadimoslem.

**Resources:** Ali Maleki.

**Software:** Mohammad Mehdi Salehi, Mobin Safarzadeh Khosrowshahi.

**Supervision:** Hossein Mashhadimoslem, Mashallah Rezakazemi, Ali A. AlHammadi, Ali Elkamel, Ali Maleki.

**Writing – original draft:** Leila Choopani, Mohammad Mehdi Salehi, Mobin Safarzadeh Khosrowshahi.

**Writing – review & editing:** Leila Choopani, Mohammad Mehdi Salehi, Mashallah Rezakazemi, Ali Elkamel, Ali Maleki.

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
