## [Decision Letter · Decision Letter 0]

16 Apr 2024

PONE-D-23-43165Removal of organic contamination from wastewater using granular activated carbon modified - Polyethylene glycol: Characterization, Kinetics and Isotherm studyPLOS ONE

Dear Dr. Rezakazemi,

Thank you for submitting your manuscript to PLOS ONE. After careful consideration, we feel that it has merit but does not fully meet PLOS ONE’s publication criteria as it currently stands. Therefore, we invite you to submit a revised version of the manuscript that addresses the points raised during the review process.

Reviewer 1:

1. The introduction is too long, it is better to reduce it to 800 words maximum.

2. Figure 1 needs reference

3. The resistance of the distilled (DI) water should be stated in the text.

4. In the results section,it is stated that "... employing GAC-PEG in the treatment of water may lead to enhanced ...", do not your test results conclusively indicate that GAC increases the removal efficiency over GAC-PEG?

5. Improve the title of Table 3. It is not appropriate to bring every specification of the table in the title

6. This manuscript has no literature review and a good literature review should be included.

7. Table 5 is not mentioned in the text, it should be described in the text.

reviewer 2:

1-You need to statement necessity of study in the introduction.

2-You need to statement aims of study in the introduction.

3- You need to statement disadvantages of adsorption in the introduction.

4- Please, transfer of sentence (The highest dye concentration

was adsorbed in pH=2 and 5 mg. L-1 tartrazine, corresponding to removal efficiencies of up to

99.81% for GAC at 0.1 g/50 mL and 90.45% for rice husk at 0.2 g/50 mL. The ideal contact time

was discovered after 120 minutes of rice husk and 60 minutes of GAC) from introduction to discussion section.

5-What criteria select of Diazinon (DZ), Amoxicillin (AMX), and Crystal Violet removal from waste water?

6-Please use same format such as : (L. (mg) -1) or (mg. L-1)

6-Which is right % or % Adsorption?

7- You need to references in the method section.

8-Add the caption above table 5

9- In Pseudo-second order unit of K2 is not (min-1) in Table 4.

10- Add QC/QA in method section.

11-You need to discussion section stronger.

12-Add recommendations and future directions for further study in the conclusion section.

13-You need to same format of references

We look forward to receiving your revised manuscript.

Kind regards,

Sara Hemati

Academic Editor

PLOS ONE

Journal Requirements:

2. Please amend the manuscript submission data (via Edit Submission) to include author Leila Choopani, Mohammad Mehdi Salehi, Hossein Mashhadimoslem, Mobin Safarzadeh Khosrowshahi, Ali A. AlHammadi, Ali Elkamel, and Ali Maleki.

Reviewers' comments:

Reviewer's Responses to Questions

**Comments to the Author**

1. Is the manuscript technically sound, and do the data support the conclusions?

Reviewer #1: Yes

Reviewer #2: Partly

2. Has the statistical analysis been performed appropriately and rigorously? 

Reviewer #1: I Don't Know

Reviewer #2: No

3. Have the authors made all data underlying the findings in their manuscript fully available?

Reviewer #1: Yes

Reviewer #2: Yes

4. Is the manuscript presented in an intelligible fashion and written in standard English?

Reviewer #1: Yes

Reviewer #2: Yes

5. Review Comments to the Author

Reviewer #1: 1. The introduction is too long, it is better to reduce it to 800 words maximum.

2. Figure 1 needs reference

3. The resistance of the distilled (DI) water should be stated in the text.

4. In the results section,it is stated that "... employing GAC-PEG in the treatment of water may lead to enhanced ...", do not your test results conclusively indicate that GAC increases the removal efficiency over GAC-PEG?

5. Improve the title of Table 3. It is not appropriate to bring every specification of the table in the title

6. This manuscript has no literature review and a good literature review should be included.

7. Table 5 is not mentioned in the text, it should be described in the text.

Reviewer #2: 1-You need to statement necessity of study in the introduction.

2-You need to statement aims of study in the introduction.

3- You need to statement disadvantages of adsorption in the introduction.

4- Please, transfer of sentence (The highest dye concentration

was adsorbed in pH=2 and 5 mg. L-1 tartrazine, corresponding to removal efficiencies of up to

99.81% for GAC at 0.1 g/50 mL and 90.45% for rice husk at 0.2 g/50 mL. The ideal contact time

was discovered after 120 minutes of rice husk and 60 minutes of GAC) from introduction to discussion section.

5-What criteria select of Diazinon (DZ), Amoxicillin (AMX), and Crystal Violet removal from waste water?

6-Please use same format such as : (L. (mg) -1) or (mg. L-1)

6-Which is right % or % Adsorption?

7- You need to references in the method section.

8-Add the caption above table 5

9- In Pseudo-second order unit of K2 is not (min-1) in Table 4.

10- Add QC/QA in method section.

11-You need to discussion section stronger.

12-Add recommendations and future directions for further study in the conclusion section.

13-You need to same format of references

6. PLOS authors have the option to publish the peer review history of their article (what does this mean?). If published, this will include your full peer review and any attached files.

Reviewer #1: **Yes: **Farideh Bagherzadeh

Reviewer #2: No

---

## [Author Response · Author response to Decision Letter 0]

11 May 2024

PONE-D-23-43165

Removal of organic contamination from wastewater using granular activated carbon modified - Polyethylene glycol: Characterization, Kinetics and Isotherm study

PLOS ONE

Dear Dr. Rezakazemi,

Thank you for submitting your manuscript to PLOS ONE. After careful consideration, we feel that it has merit but does not fully meet PLOS ONE’s publication criteria as it currently stands. Therefore, we invite you to submit a revised version of the manuscript that addresses the points raised during the review process.

Dear Dr. Sara Hemati

We thank the reviewers and the editor for their interesting and helpful comments. Below, we give our responses to all the specific issues and indicate the associated changes we have made to the manuscript. We earnestly appreciate the views, comments, and suggestions provided by all the reviewers. We have revised and attempted to improve the manuscript accordingly. Revisions are highlighted in the revised manuscript.

Reviewer 1:

1. The introduction is too long, it is better to reduce it to 800 words maximum.

Response: Thank you so much for your respectful comments. This important topic, as kindly highlighted by the reviewer, was considered in the manuscript. We simplified the introduction section, pp.2-4, lines 8-12, 15-19, 20-22, 1-3, 9-14 & 12-16 as follows:

” Clean water is essential for human health and the ecosystem. In many parts of the world, access to clean and safe water is a significant challenge. In order to be clean and safe, water must be free of all pollutants and harmful factors. Pollutants include chemical, physical, and biological threats to health. Diazinon (DZ), known as dimpylate, is an organophosphorus pesticide (OPPs) frequently employed in urban and agricultural settings.” 

“Antibiotics are considered essential medical medicines, but their release into the environment and aquatic resources has had adverse effects even at modest doses. Semi-synthetic penicillin known as amoxicillin (AMX), among the most popular in Europe and other parts of the world, has been used alone and in combination with the beta-lactamase inhibitor clavulanic acid since the 1970s.” “Crystal Violet (CV) is a toxin dye that, according to research, can cause skin irritation and ocular problems in addition to the ability to cause cancer in mammalian cells. The chemical structures of DZ, AMX, and CV are illustrated in Fig.1.”

Fig.1. Chemical structure of (a) DZ, (b) AMX, and (C) CV.

It is essential to remove such organic contaminants from contaminated water to preserve the environment and the general public's health.” 

“However, among the disadvantages of the adsorption process, we can point out the difficult recovery, inefficient adsorption method, and weak interactions between the adsorbent surface and the adsorbed surface. In this method several adsorbents are used, one of which is carbon-based adsorbents. Carbon-based adsorbents appropriate for gas adsorption include CMS, granular activated carbon (GAC), graphene, carbon nanotubes, and fullerenes”.

“A maximum adsorption capacity was reported for DZ, AMX, and CV adsorption. A Langmuir and Freundlich isotherm were investigated. A pseudo-first-order and a pseudo-second-order model was developed to describe the adsorption kinetics. As a result of the experiments, DZ, AMX, and CV were able to achieve their maximum adsorption capability using GAC-PEG.”

2. Figure 1 needs reference

Response: Thank you for your comment. Changes have been done in revised manuscript. This important topic is now considered and revised the mentioned (Fig. 1, p.3) in the amended manuscript. Figure 1 was drawn by the authors as well.

3. The resistance of the distilled (DI) water should be stated in the text.

Response: Thanks for your attention. The change has been done in revised manuscript. We consider this comment and mentioned in p.5, line 7.

4. In the results section, it is stated that "... employing GAC-PEG in the treatment of water may lead to enhanced ...", do not your test results conclusively indicate that GAC increases the removal efficiency over GAC-PEG?

Response: Thank you for your comment. Based on the results of the testing of the GAC adsorption capacity for the three organic contaminants (Diazinon (DZ) = 516.21 mg. g-1, Amoxicillin (AMX) = 564.87 mg. g-1, and Crystal Violet (CV) = 438.32 mg. g-1), Table 5 of the revised manuscript illustrates the results. Table 5 in the revised manuscript shows that GAC-PEG has a higher adsorption capacity than GAC due to its synergistic effects.

To produce the final adsorbent, GAC and PEG combine to create a synergistic effect. Consequently, the adsorbent's sorption capacity is further enhanced by cooperative interactions. In this adsorbent, several components contribute unique properties to the adsorption process that complement each other and result in synergistic effects. Having a high surface area and a mesoporous structure, GAC-PEG provides a perfect environment for adsorption. As a result of its complementary properties, this material outperforms its components when it comes to adsorption efficiency. Another reason is the increase in functional groups. Modified GAC contains some of these peaks, indicating that PEG tends to make it hydrophilic. During the process of functionalization, it seems that chain breaks gradually increase surface functional groups [1-3].

[1] Salehi, Mohammad Mehdi, Fereshte Hassanzadeh-Afruzi, Golnaz Heidari, Ali Maleki, and Ehsan Nazarzadeh Zare. “In situ preparation of MOF-199 into the carrageenan-grafted-polyacrylamide@ Fe3O4 matrix for enhanced adsorption of levofloxacin and cefixime antibiotics from water.” Environmental Research 233 (2023): 116466. 

[2] Saeed, Ahmed M. "Temperature effect on swelling properties of commercial polyacrylic acid hydrogel beads." Int. J. Adv. Biol. Biomed. Res 1, no. 12 (2013): 1614-1627. 

[3] Loyez, Médéric, Maxwell Adolphson, Jie Liao, Sanskar Thakur, and Lan Yang. "pH-sensitive optical micro-resonator based on PAA/PVA gel swelling." In Optics and Biophotonics in Low-Resource Settings IX, vol. 12369, pp. 36-39. SPIE, 2023.

5. Improve the title of Table 3. It is not appropriate to bring every specification of the table in the title. 

Response: Thank you so much for your comment. This important topic, as kindly highlighted by the reviewer, is now considered in the amended manuscript, in p.10 line 8. The new title is Specific textural attributes of GAC-Pure and GAC-PEG samples.

6. This manuscript has no literature review and a good literature review should be included.

Response: Thank you so much for your comment. This important topic, as kindly highlighted by the reviewer, is now considered in the amended manuscript. We added the literature review in the introduction section, pp.2-4, lines 8-12, 15-19, 20-22, 1-3, & 9-14 as follows:

” Clean water is essential for human health and the ecosystem. In many parts of the world, access to clean and safe water is a significant challenge. To be clean and safe, water must be free of all pollutants and harmful factors [1]. Pollutants include chemical, physical, and biological threats to health. Diazinon (DZ), known as dimpylate, is an organophosphorus pesticide (OPPs) frequently employed in urban and agricultural settings.” 

“Antibiotics are considered essential medical medicines, but their release into the environment and aquatic resources has had adverse effects even at modest doses [4]. Semi-synthetic penicillin, known as amoxicillin (AMX), among the most popular in Europe and other parts of the world, has been used alone and in combination with the beta-lactamase inhibitor clavulanic acid since the 1970s[5].”

“Crystal Violet (CV) is a toxin dye that, according to research, can cause skin irritation and ocular problems in addition to the ability to cause cancer in mammalian cells [7]. The chemical structures of DZ, AMX, and CV are illustrated in Fig. 1. [8-10].” 

“Removing such organic contaminants from contaminated water is essential to preserve the environment and the general public's health”.

“However, among the disadvantages of the adsorption process, we can point out the difficult recovery, inefficient adsorption method, and weak interactions between the adsorbent surface and the adsorbed surface [13]. In this method, several adsorbents are used, including carbon-based adsorbents. Carbon-based adsorbents appropriate for gas adsorption include CMS, granular activated carbon (GAC) , graphene , carbon nanotubes , and Graphitic Carbon Nitride [14-16].”

7. Table 5 is not mentioned in the text, it should be described in the text.

Response: Thank you so much for your comment. This important topic, as kindly highlighted by the reviewer, is now considered in the amended manuscript. We added the literature review in the section 3.3., pp.15-16, lines 24-26 & 1-4, as follows:

“Table 5 compares the maximal DZ, AMX, and CV adsorptive capability of the GAC-PEG to those of other adsorbents mentioned in earlier studies. Compared to several other adsorbents that have been previously reported, the created GAC-PEG adsorbent showed high Qmax. The different physicochemical properties of the made adsorbent systems, including a huge surface area and plenty of reactive adsorption sites, such as hydroxyl groups covering the surface, effectively adsorb OPPs, antibiotics, and toxic dyes from wastewater pollutants.”

Reviewer 2:

1-You need to statement necessity of study in the introduction.

Response: Thank you so much for your respectful comments. This important topic, as kindly highlighted by the reviewer, was considered in the manuscript. Changes have been made to the Introduction section of the revised manuscript. We mentioned in sec. 1., pp.2-4, lines 8-22 as follows:

” Clean water is essential for human health and the ecosystem. In many parts of the world, access to clean and safe water is a significant challenge. To be clean and safe, water must be free of all pollutants and harmful factors [1]. Pollutants include chemical, physical, and biological threats to health. Diazinon (DZ), known as dimpylate, is an organophosphorus pesticide (OPPs) frequently employed in urban and agricultural settings.” 

“Antibiotics are considered essential medical medicines, but their release into the environment and aquatic resources has had adverse effects even at modest doses [4]. Semi-synthetic penicillin, known as amoxicillin (AMX), among the most popular in Europe and other parts of the world, has been used alone and in combination with the beta-lactamase inhibitor clavulanic acid since the 1970s [5].”

“Crystal Violet (CV) is a toxin dye that, according to research, can cause skin irritation and ocular problems in addition to the ability to cause cancer in mammalian cells [7]. The chemical structures of DZ, AMX, and CV are illustrated in Fig. 1. [8-10]” 

“Removing such organic contaminants from contaminated water is essential to preserve the environment and the general public's health”.

2-You need to statement aims of study in the introduction.

Response: Thank you so much for your respectful comments. This important topic, as kindly highlighted by the reviewer, was considered in the manuscript. We consider the aims of study in the introduction section, p. 3, lines 9-14 as follows:

“However, among the disadvantages of the adsorption process, we can point out the difficult recovery, inefficient adsorption method, and weak interactions between the adsorbent surface and the adsorbed surface [13]. In this method, several adsorbents are used, including carbon-based adsorbents. Carbon-based adsorbents appropriate for gas adsorption include CMS, granular activated carbon (GAC), graphene, carbon nanotubes, and Graphitic Carbon Nitride [14-16].”

3- You need to statement disadvantages of adsorption in the introduction.

Response: Thank you so much for your respectful comments. This important topic, as kindly highlighted by the reviewer, was considered in the manuscript. We consider the aims of study in the introduction section, p. 3, lines 13-18 as follows:

“However, among the disadvantages of the adsorption process, we can point out the difficult recovery, inefficient adsorption method, and weak interactions between the adsorbent surface and the adsorbed surface [13]. In this method, several adsorbents are used, including carbon-based adsorbents. Carbon-based adsorbents appropriate for gas adsorption include CMS, granular activated carbon (GAC), graphene, carbon nanotubes (CNTs), and Graphitic Carbon Nitride [14-16].”

4- Please, transfer of sentence (The highest dye concentration was adsorbed in pH=2 and 5 mg. L-1 tartrazine, corresponding to removal efficiencies of up to 99.81% for GAC at 0.1 g/50 mL and 90.45% for rice husk at 0.2 g/50 mL. The ideal contact time was discovered after 120 minutes of rice husk and 60 minutes of GAC) from introduction to discussion section.

Response: Thanks for the good comment. Thank you for your admirable attention. This text is related to another researcher's work, which is included in the introduction section to make it more complete and express the concept. We mentioned before this sentence “Khader et al. [22] used rice husk as a biomass-derived adsorbent and commercial GAC to separate the tartrazine color from an aqueous solution and compared the two.”

[22] Khader, Eman H., Thamer J. Mohammed, and Talib M. Albayati. "Comparative performance between rice husk and granular activated carbon for the removal of azo tartrazine dye from aqueous solution." Desalin Water Treat 229 (2021): 372-383.

5-What criteria select of Diazinon (DZ), Amoxicillin (AMX), and Crystal Violet removal from waste water?

Response: Thank you for your comment. Today, pharmaceutical pollution, dyes, insecticides, and pesticides have a significant potential for the environment due to their high consumption, and in particular pharmaceutical compounds that affect human and animal health. Medicines through the pharmaceutical industry, wastewater hospitals, human and animal excrement, the output of urban treatment plants, and the discharge of waste in poultry breeding, and industry enter the water and soil. Diazinon (DZ) is an organophosphorus insecticide frequently employed in urban and agricultural settings. High residual DZ concentrations were also discovered in urban waterways and wastewater treatment plant effluents. The problem that antibiotics such as amoxicillin (AMX) cause even in low concentrations in aquatic and terrestrial ecosystems is bacterial resistance and increased allergic reactions. Crystal Violet (CV) is a toxin dye that, according to research, can cause skin irritation and ocular problems in addition to the ability to cause cancer in mammalian cells Also, organic dyes demonstrate toxicity and low biodegradability, with carcinogenic effects on aquatics because of their stability and structural aromaticity. According to the mentioned risks for the environment and human health, water treatment from pesticides, organic dye, and pharmaceutical pollution contaminants is vital [1-3].

[1] Salehi, Mohammad Mehdi, Fereshte Hassanzadeh-Afruzi, Farhad Esmailzadeh, Leila Choopani, Kimiya Rajabi, Hosein Naeimy Kuzekanan, Mojtaba Azizi, Oleg M. Demchuk, and Ali Maleki. "Chlorpyrifos and diazinon elimination through pAAm-g-XG/HKUST-1@ Fe3O4 biopolymer nanoadsorbent hydrogel from wastewater: Preparation, characterization, kinetics and isotherm." Separation and Purification Technology 334 (2024): 126097.

[2] Beigi, Paria, Fatemeh Ganjali, Fereshte Hassanzadeh-Afruzi, Mohammad Mehdi Salehi, and Ali Maleki. "Enhancement of adsorption efficiency of crystal violet and chlorpyrifos onto pectin hydrogel@ Fe3O4-bentonite as a versatile nanoadsorbent." Scientific Reports 13, no. 1 (2023): 10764.

[3] Mosavi, Seyedeh Soghra, Ehsan Nazarzadeh Zare, Hossein Behniafar, and Mahmood Tajbakhsh. "Removal of amoxicillin antibiotic from polluted water by a magnetic bionanocomposite based on carboxymethyl tragacanth gum-grafted-polyaniline." Water 15, no. 1 (2023): 202.

6-Please use same format such as: (L. (mg) -1) or (mg. L-1)

Response: Thanks for the good comment. All these have been implemented in the amended manuscript. The mentioned comments are unified as yellow color in all sections of the manuscript.

6-Which i

---

## [Editor Report · Decision Letter 1]

16 May 2024

Removal of organic contamination from wastewater using granular activated carbon modified - Polyethylene glycol: Characterization, Kinetics and Isotherm study

PONE-D-23-43165R1

Dear Dr. Choopani,

We’re pleased to inform you that your manuscript has been judged scientifically suitable for publication and will be formally accepted for publication once it meets all outstanding technical requirements.

Kind regards,

Sara Hemati

Academic Editor

PLOS ONE

Additional Editor Comments (optional):

Accept

Reviewers' comments:

No comment

---

## [Editor Report · Acceptance letter]

4 Jun 2024

PONE-D-23-43165R1 

PLOS ONE

Dear Dr. Rezakazemi, 

I'm pleased to inform you that your manuscript has been deemed suitable for publication in PLOS ONE. Congratulations! Your manuscript is now being handed over to our production team.

Kind regards, 

on behalf of

Dr. Sara Hemati 

Academic Editor

PLOS ONE